# Who is Speaking? Speaker-Aware Multiparty Dialogue Act Classification

**Ayesha Qamar**[*], **Adarsh Pyarelal**[†] and **Ruihong Huang**[*]

[*]Texas A&M University, [†]University of Arizona

{ayesha, huangrh}@tamu.edu, adarsh@arizona.edu

## Abstract

Utterances do not occur in isolation in dialogues; it is essential to have the information of who the speaker of an utterance is to be able to recover the speaker's intention with respect to the surrounding context. Beyond simply capturing speaker switches, identifying how speakers interact with each other in a dialogue is crucial to understanding conversational flow. This becomes increasingly important and simultaneously difficult to model when more than two interlocutors take part in a conversation. To overcome this challenge, we propose to explicitly add speaker awareness to each utterance representation. To that end, we use a graph neural network to model how each speaker is behaving within the local context of a conversation. The speaker representations learned this way are then used to update their respective utterance representations. We experiment with both multiparticipant and dyadic conversations on the MRDA and SwDA datasets and show the effectiveness of our approach.[1]

## 1 Introduction

Dialogue acts (DAs) describe the intention behind a speaker's utterance in a conversational context (Searle, 1969). Recognizing DAs is helpful for several downstream tasks such as question-answering (Goo and Chen, 2018a), summarization (Goo and Chen, 2018b; Oya and Carenini, 2014), and conversational AI (Xu et al., 2018; Wang et al., 2020).

Utterances are the building blocks of a conversation, but they do not occur in isolation. To recover the communicative function behind an utterance, we need to consider its *context*. The context here refers not just to previous utterances, but also to dialogue structure, speaker behavior, etc. Utterances are multifunctional in the sense that they encode many roles—e.g., turn management, relaying communicative intentions, etc. (Bunt, 2006).

---

[1]Code available at https://github.com/Ayesha55/Speaker-graph-model-for-DAs-EMNLP

| Speaker | Utterance | Label |
|---------|-----------|-------|
| A | and i've been sharing them with u u w folks also. | Statement |
| B | i'm sorry you've been what? | Understanding check |
| C | okay. | Backchannel |
| B | showing them? | Understanding check |
| A | sharing them with the u w folks. | Correct Misspeaking |
| B | sharing them. | Mimic |
| B | okay. | Acknowledgement |
| B | okay. | Backchannel |
| C | okay. | Floor Grabber |

Table 1: A short excerpt from a dialogue involving multiple speakers from the MRDA corpus.

One example of this is the broader category of the *Question* DA, which, along with seeking information, also implicitly gives the floor to the addressee. Therefore, interlocutor information is needed to instill situational awareness of speakers in a conversation. In a dialogue, speakers influence each other which in turn dictates how they behave and convey their intentions. Different speakers can also fulfill different roles, for example, within the local context of a conversation, a speaker can be leading the conversation by providing information while others could be active listeners and signal they are following along by means of *backchannels* as shown in Table 1.

There has not been much prior work on incorporating speaker awareness for DA classification, particularly in the case of multiparty conversations (i.e., conversations involving more than two interlocutors). While there is some research on modeling speakers for DA classification, it is mostly focused on two-party conversations. Chi et al. (2017) use a separate BiLSTM to model each speaker in a dyadic setting. Although this is one way to capture speaker behavior, it does not scale to the case of multiparty

dialogues and real-world situations where the number of speakers can vary at test time. Recently, He et al. (2021) introduced a way to capture speaker turns by learning *turn* embeddings. For this to work in a multiparty setting, the dialogue needs to be reduced to a dyadic one. This is not ideal since doing this would take away crucial information on different participants' contributions.

In the related field of emotion recognition in dialogues (ERC), some work has been done to encode speakers (Lee and Choi, 2021; Ghosal et al., 2019; Zhang et al., 2019; Song et al., 2023). These works primarily use graph neural networks (GNNs) to enhance the representation of utterances. Speaker information is injected either through edges or by adding speaker nodes to the graph. Since the main objective of these methods is to encode utterances better, often different types of edges are introduced to better capture the different relationships between utterances. For example, different types of edges for past and future connections (Ishiwatari et al., 2020) or learning dedicated edge types for each individual speaker (Ghosal et al., 2019). This translates to learning a bigger model with more parameters as well as a denser graph with many edges. Due to the large memory requirements of such models, training them can be hard in a resource-scarce situation. We show in our work that instead of using the speaker as a means to capture utterance context better, directly learning the speaker representations is more efficient. This results in a simpler graph with fewer edges.

In this work, we present a GNN-based framework that takes contextual utterance vectors as input and encodes conversation dynamics by connecting the node of each speaker with their utterances. The learned speaker representations are then concatenated with the utterance representations to get *speaker-enriched* utterance representations which are then used for DA classification. We conduct extensive analysis on the ICSI Meeting Recorder Dialog Act (MRDA) corpus (Shriberg et al., 2004) to understand how the fine-grained DA classes are affected by incorporating speaker representations.

## 2 Related Work

**DA classification** Most existing works on DA classification (Kumar et al., 2018; Chen et al., 2018; Raheja and Tetreault, 2019; Bothe et al., 2018; Khanpour et al., 2016) use recurrent neural networks (RNNs) as backbone models to represent utterances. To capture context, a hierarchical RNN with some variation is used. DA label sequences can help learn associations between tags that occur together or follow a pattern; conditional random fields (CRFs) have extensively been used for this purpose. Additionally, Raheja and Tetreault (2019) use attention to leverage context more effectively.

To make utterances speaker turn-aware, He et al. (2021) learn two turn embeddings to capture turns. The embedding vector based on the speaker label is added to the utterance vector and then passed to another RNN to capture context. Whereas Shang et al. (2020) use a CRF layer to model turn-taking. While this is an effective way to inject speaker information, multiparty dialogues need to be converted to dyadic dialogues for these approaches to work.

Chi et al. (2017) work in the setting of conversational agents and aim to capture the speaker roles of the agent and the user using a separate BiLSTM for each. This technique cannot be applied to natural conversations where the number of participants is not fixed.

Colombo et al. (2020) use three RNN encoders: word-level, speaker-level, and utterance-level. The speaker-level RNN is fed utterances grouped by speaker to capture speaker personas. The representations from this encoder are then fed to the utterance-level encoder for wider context modeling. Their work differs from ours in the way that they model each speaker. While they encode utterance context at the speaker level, we learn an explicit speaker representation.

**Graph-based methods to encode speakers in dialogues** We discuss some of the work here that uses GNNs (Scarselli et al., 2008) for emotion recognition in conversations. Ghosal et al. (2019) build a graph with only utterances as nodes and incorporate context at the speaker level using different edges. There is a separate directed edge type from each speaker to every other speaker while also differentiating the directionality of past and future. Thus, there are $2M^2$ edge types in total, where $M$ is the number of unique speakers. This technique does not scale in cases where there are more speakers in a dialogue at test time than the maximum number of speakers seen during training.

Shen et al. (2021b) and Sheng et al. (2020) construct a dialogue graph by only using utterance nodes. Two different edge types are used to indicate if two connected utterances share the same speaker or not. In some ways, this is akin to converting a

multiparty dialogue to a two-party one, since the utterances only have binary information of speakers.

Some other works construct the graph with both speaker and utterance nodes. This includes Lee and Choi (2021), who treat ERC as a relation extraction task through a GNN. While Zhang et al. (2019) share speaker nodes across dialogues, we are more interested in capturing the dynamics of speaker influence within a local context and assume that any new speaker can be seen during test time. Liang et al. (2021) learn fixed embeddings for each unique speaker and use them to initialize the speaker nodes.

One thing common to all these papers is that they explicitly learn utterance node representations through the graph (Sun et al., 2021; Song et al., 2023)—i.e., they build the graph representations as a means to get better utterance representations. The speaker nodes serve as global nodes for effective message passing between utterances. This not only results in a very dense graph but also increases other complexities by means of introducing different edge types based on speaker associations. In contrast, we are interested in learning speaker representations and how using those can enrich the utterances. Doing so only requires connecting the speaker to its utterance nodes. We present a simple graph construction scheme and show that modeling speakers directly is not only efficient but effective as well. Our proposed approach can be used on top of any pre-trained utterance encoding system.

## 3 Task

A dialogue $\mathcal{D}$ with $|\mathcal{P}|$ participants can be defined as a collection of utterances $\mathcal{D} = \{u_0, u_1, ..., u_l\}$. That is, $|\mathcal{D}| = l$. Each utterance is associated with a speaker given by $S(u_i) = s_i$ and a DA label $y_i$. The aim of DA classification is to assign a DA label from a set of labels $C$ to each utterance $u_i$ in $\mathcal{D}$.

## 4 Model

In this section, we describe the components of our model, an overview of which is presented in Figure 1.

### 4.1 Utterance Encoder

The utterance encoder is the same baseline as presented in He et al. (2021).[2] The sentence separator  token from the last layer of the RoBERTa[3] Liu

---

[2] We include the base model without the speaker turn embeddings.
[3] https://huggingface.co/docs/transformers/model_doc/roberta

et al. (2019) model is used to derive a representation $e(u_i)$ for each utterance. We use RoBERTa-base with 12 layers and fine-tune the final layer. To get the contextual representations, the utterances are passed to a bidirectional GRU (Cho et al., 2014). These final utterance representations are defined by

$$[h_0, h_1, .., h_l] = \text{GRU}\left([e(u_0), e(u_1), ..e(u_l)]\right),\tag{1}$$

where $h_i \in \mathbb{R}^d$. Here, $d$ is the dimension of each utterance vector.

### 4.2 Speaker Turn Indicating Tokens

Participants speak intermittently in a dialogue. If a single speaker has had the floor for the past few utterances and the current utterance is also by the same speaker, then the chances of certain DA labels such as *backchannel, mimic, and collaborative completion* decrease, whereas the chances for other DA labels such as *repeat, self-correct misspeaking* increase. This example shows why modeling speaker turns is crucial.

To capture this behavior of turn-taking at the utterance level, a turn indication token can be used (Żelasko et al., 2021). Each utterance is prepended with a special token to get the updated utterance $u'$ as given by (2). These updated utterances are then fed to the encoder defined in § 4.1. The upside of encoding speaker turns this way is that no new parameters are introduced into the model.

$$u'_i = \begin{cases} \langle\text{same}\rangle + u_i, & \text{if } S(u_i) = S(u_{i-1}) \\ \langle\text{switch}\rangle + u_i, & \text{if } S(u_i) \neq S(u_{i-1}) \end{cases}\tag{2}$$

### 4.3 Graph Speaker Modeling

The addition of the *turn* token can still only capture the binary speaker transitions. While this might be enough in the case of a dyadic conversation, it reduces a multiparty conversation to a two-party one. To overcome this, on top of the *turn* tokens, we also learn a graph-based representation for each speaker and use it to inform each utterance of its speaker.

**Graph Structure** We define the graph as $\mathcal{G} = \langle \mathcal{V}, \mathcal{E}, \mathcal{R} \rangle$, where the nodes $v_i \in \mathcal{V}$ can be one of two types: an utterance $v_i^u$ or a speaker $v_i^s$. Labeled edges $e_{ij} \in \mathcal{E}$ denote edges between $v_i$ and $v_j$. Finally, $r \in \mathcal{R}$ is the type of relation an edge represents.

We introduce a single type of relation $\mathcal{R}$ in the graph that an edge $e_{ij}$ connecting two nodes $v_i$

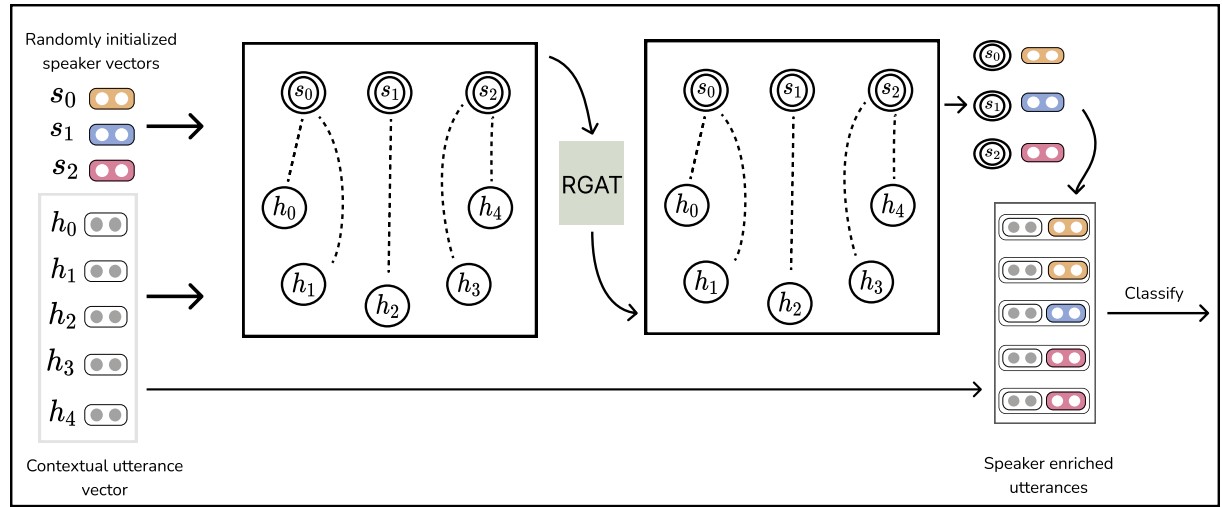

Figure 1: Example graph for a dialogue with five utterances and three unique speakers. Dotted lines denote edges of type $(v_i^u, v_{S(u_i)}^s)$.

and $v_j$ can take. We denote it by $(v_i^u, v_{S(u_i)}^s)$, it represents an undirected edge between an utterance and its speaker. We show in § 6.3 that compared to more complex graph structures, this one type of edge is enough to model speakers effectively.

**Graph Construction** Given a dialogue $\mathcal{D}$ with $|\mathcal{P}|$ speakers, utterance nodes are initialized using (1), following Zhang et al. (2019), and speaker nodes are randomly initialized. Therefore, the total number of nodes is $|\mathcal{V}| = |\mathcal{D}| + |\mathcal{P}|$.

**Speaker Learning** The utterance encoder in all speaker graph experiments is initialized with the trained weights of the encoder given in § 5.3. The $W_{\text{baseline}}$ layer is removed, and the encoder weights are kept fixed during training. We use a Relational Graph Attention Network (RGAT) (Busbridge et al., 2019) to model the speakers with respect to their utterances. The graph is constructed and initialized as detailed above and passed to RGAT to get updated representations of the vertices given by:

$$\widetilde{\mathcal{V}} = \text{RGAT}(\mathcal{V}, \mathcal{E}, \mathcal{R}) \qquad (3)$$

**DA Classification** The updated speaker representations $\widetilde{v}_i^s \in \widetilde{\mathcal{V}}$ from the graph are extracted and concatenated with their respective utterance representation $h_i$ from (1). The final utterance representation is given by:

$$h_i^{\text{speaker-enriched}} = [h_i; \widetilde{v}_{S(u_i)}^s] \qquad (4)$$

Here, $\widetilde{v}_{S(u_i)}^s$ is the graph node representation of $ith$ utterance's speaker, $h^{\text{speaker-enriched}} \in \mathbb{R}^{2d}$. Finally,

| Dataset | $|C|$ | $|\mathcal{P}|$ | Dialogues | | | Utterances | | |
|---|---|---|---|---|---|---|---|---|
| | | | Train | Val | Test | Train | Val | Test |
| MRDA | 50 | 3–9 | 50 | 11 | 11 | 75k | 15.3k | 15k |
| SwDA | 43 | 2 | 1003 | 112 | 19 | 193.3k | 20.2k | 4.5k |

Table 2: Number of classes ($|C|$), participants per dialogue ($|\mathcal{P}|$), and number of dialogues and utterances in each split.

$h^{\text{speaker-enriched}}$ is passed to a feed-forward network to get the predicted label:

$$\hat{y} = W_{\text{Spk-Graph}}\left(h^{\text{speaker-enriched}}\right). \qquad (5)$$

## 5 Experiments

### 5.1 Dataset

We experiment with two publicly available datasets for DA classification. For both the datasets, we use the dataset split used by Lee and Dernoncourt (2016). The dataset overview and distribution of utterances are given in Table 2.

**MRDA** The first dataset is the MRDA corpus, which consists of around 72 hours of natural conversations, with an average dialogue length of 1445.41 utterances. There are 11 general and 39 specific tags, along with three types of disruptions and a non-speech label. The MRDA corpus follows a hierarchical annotation scheme, where each utterance is labeled with compulsory general and need-based zero or more specific tags.

Over the years, many grouping schemes have been devised to consolidate the MRDA tags into

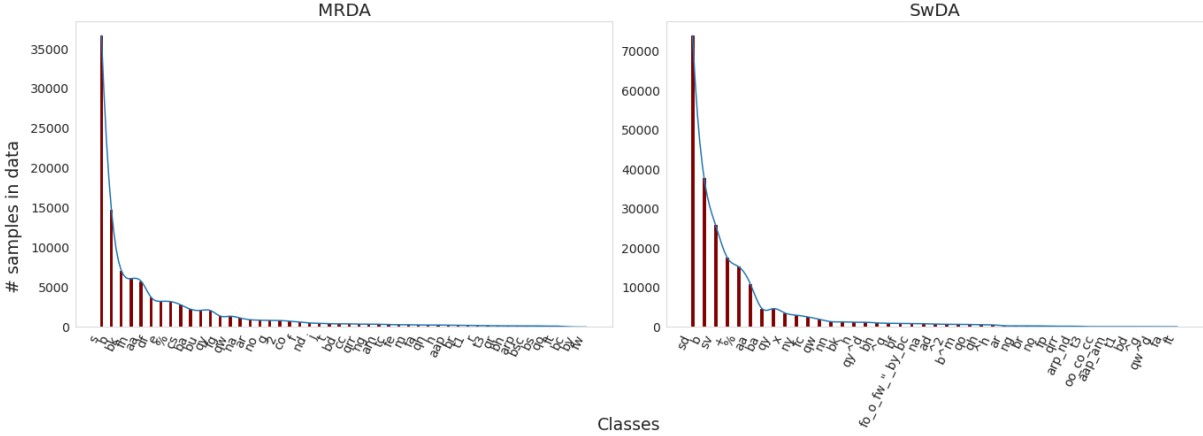

Figure 2: Both the datasets follow a long-tailed distribution where a few frequent classes have a very large number of samples and there are many infrequent classes with only a few samples.

higher-level categories. One of the most widely used such grouping is the basic-tags introduced in Ang et al. (2005)[4]. While the majority of the existing works using MRDA focus on these five coarse-grained categories (Kumar et al., 2018; Chen et al., 2018; Raheja and Tetreault, 2019; Bothe et al., 2018; Khanpour et al., 2016; He et al., 2021), we focus on the fine-grained labels.

In particular, an utterance is always assigned the first (in case of more than one) specific tag if one is present. If the DA label consists of only a general tag, then that label is used.[5] We made three changes to the labels: (i) we dropped *rising tone* since it is not a DA (Dhillon et al., 2004), (ii) we dropped *declarative question* because it captures the syntactic structure of the utterance, it is used when a question is framed as a statement, and (iii) when a pipe symbol (|) is used to annotate the floor mechanism at the start of an utterance, we take the label assigned to the later part of the utterance instead of the floor mechanism. Floor mechanisms at the start of an utterance are not the most informative tag when only choosing one label. Examples of annotated utterances are shown in Table 1.

**SwDA** The second dataset we work with is the Switchboard Dialog Act corpus (Jurafsky, 1997), a collection of telephone conversations between two people on a pre-specified topic. The SwDA [6] corpus contains 43 DA types, and the average dialogue

length is 192.3 utterances.

## 5.2 Evaluation Metric

The distributions of DAs in both the MRDA and the SwDA datasets are highly skewed, with the five most frequent classes making up 66.7% and 78.1% of the data in MRDA and SwDA respectively. Deep learning models have been known to be biased toward the classes with the most number of samples in the data (Johnson and Khoshgoftaar, 2019). As shown in Figure 2, both datasets follow a long-tailed distribution, with a few majority classes accounting for most of the data. Previous works primarily report the accuracy of their models.

However, relying solely on accuracy to judge a model on a highly imbalanced dataset can be problematic (Gu et al., 2009; Chawla, 2009; Bekkar et al., 2013). Model performance can be overestimated by accuracy even if it only performs well on a few frequent classes (Kotsiantis et al., 2006). Building a system that can also perform well on minority classes for DAs is important for many downstream tasks. For example, infrequent DAs such as *repetition request* and *partial reject* can be effective indicators of misunderstanding for conversational agents (Aberdeen and Ferro, 2003). DAs such as *command* and *suggestion* that together make up less than 4% of data in MRDA, are especially important for conversational AI assistants to recognize.

The F1 score is the harmonic mean of precision and recall and it serves as a balanced assessment of both these measures (Buckland and Gey, 1994). With the end goal of building a system that can be useful for several downstream tasks, we report the macro F1, precision, and recall scores to give equal

---

[4]These are Statement, Question, Floorgrabber, Backchannel, and Disruption

[5]We use the full labels as presented here: `https://github.com/NathanDuran/MRDA-Corpus`

[6]`https://github.com/cgpotts/swda`

weight to each class. Although accuracy is not the most appropriate metric to use in this setting, we still report it for a sense of comparison with prior work.

### 5.3 Baseline

As a baseline, we take the utterance representations from the encoder presented in § 4.1 and pass them to a feed-forward layer to get class probabilities:

$$\hat{y}_{\text{baseline}} = W_{\text{baseline}}(h). \qquad (6)$$

We use a fixed chunk size of 128 for all the baselines. Based on the results in He et al. (2021), this is the best chunk size for SwDA. The effect of chunk size on DA classification accuracy is negligible for the MRDA dataset. We compare the speaker graph model with two baselines. The other, *turn-aware* baseline, has augmented turn-aware input as presented in § 4.2.

### 5.4 Experimental Setup

The implementation is done in PyTorch (Paszke et al., 2019) version 1.12.1+cu113 and PyTorch-geometric (Fey and Lenssen, 2019) version 2.1.0 was used for the graph experiments. The Hugging-face transformers library was used for RoBERTa. The AdamW (Loshchilov and Hutter, 2019) optimizer was used with a weight decay of $5e-4$. The LR is set to $1e-4$ and $1e-3$ for the baseline and the graph models respectively. The test scores are taken from the epoch where the model gets the best validation score. All models are trained using the cross-entropy loss. Because of memory constraints, we segment each dialogue into non-overlapping chunks of size 128 in all experiments unless otherwise stated. Further details can be found in Appendix A.

## 6 Results and Discussion

We compare the speaker graph model against the stronger of the two baselines and consequently also use the turn-aware input for the graph model. All the results are an average of 5 runs to account for the fluctuation introduced by randomness. The speaker graph model's performance is statistically significant with p<0.0001 using Student's t-test.

### 6.1 Prior Work Baselines

As mentioned previously, prior work has focused on the high-level categories of DA and they report accuracy only. Table 3 shows the results on fine-grained classes for the presented speaker graph model along with several related systems. Our results differ from previous work in that we report the accuracy from the epoch with the best F1 score on the validation set instead of the best accuracy.

Most papers on DA classification do not have their code publicly available (with the exception of He et al. (2021)), making comparisons with them difficult. We present the results for three systems from DA classification.

- BiLSTM+CRF (Kumar et al., 2018) is a commonly used model in DA classification. This is a hierarchical BiLSTM model with a CRF layer on top.
- BiLSTM SelfAtt-CRF (Raheja and Tetreault, 2019) is similar to BiLSTM+CRF, but they also introduce an attention component to effectively incorporate contextual utterance representation from previous timestamp $t_i - 1$ and word representations from $t_i$ to compute utterance representation for $t_i$.
- Turn Modeling (He et al., 2021) learns two speaker switch embeddings that are added to utterance embeddings. The utterance encoder is the same as our baseline.

We also compare with related works from emotion recognition,

- DialogueGCN (Ghosal et al., 2019) leverages self and inter-speaker dependency to model the entire conversational context using a graph neural network.
- DialogueRNN (Majumder et al., 2019) keeps track of the individual party states throughout the conversation using speaker-specific RNN.
- DialogXL (Shen et al., 2021a) uses XLNet Yang et al. (2019) as the base model and incorporates dialogue and speaker-level self-attention.
- DAG-ERC (Shen et al., 2021b) treats the conversation as a directed acyclic graph with only utterance node. Two different edge types indicate if two connected utterances belong to the same speaker or not.
- DialogCRN (Hu et al., 2021) uses separate LSTMs to model speaker and context dependencies.
- SUNET (Song et al., 2023) construct a graph with both speaker and utterance nodes. They use a GRU to update speaker nodes based on their representations from previous layers.

| Model | MRDA | | | | SwDA | | | |
|---|---|---|---|---|---|---|---|---|
| | Precision | Recall | F1 | Accuracy | Precision | Recall | F1 | Accuracy |
| RoBERTa$_{base}$ | 37.14 | 39.58 | 36.33 | 64.24 | 47.12 | 51.24 | 46.65 | 72.22 |
| BiLSTM+CRF♣ | 36.09 | 32.87 | 32.69 | 65.38 | 59.11 | 53.69 | 54.91 | 79.1 |
| BiLSTM SelfAtt+CRF♢ | 34.32 | 32.19 | 31.21 | 63.66 | 54.41 | 51.14 | 51.07 | 74.44 |
| Turn Modeling♣ | 43.52 | 38.92 | 38.77 | 67.0 | 62.48 | 56.9 | 57.96 | 81.0 |
| DialogueGCN$_{RoBERTa}$♣ | 40.18 | 35.37 | 34.81 | 61.18 | 59.55 | 54.11 | 55.34 | 79.71 |
| DialogueRNN$_{RoBERTa}$♣ | 37.58 | 38.9 | 37.12 | 65.04 | 56.04 | 61.87 | 57.21 | 81.2 |
| DialogXL♣ | 40.67 | 37.99 | 37.97 | 63.79 | 59.46 | 57.7 | 56.80 | 79.68 |
| DAG-ERC♣ | 43.64 | 38.46 | 39.34 | 67.01 | 59.24 | 54.6 | 55.77 | 78.62 |
| DialogCRN$_{RoBERTa}$♣ | 40.43 | 37.78 | 37.58 | 65.05 | 58.18 | 54.51 | 54.91 | 77.04 |
| SUNET♢ | 37.66 | 40.72 | 38.11 | 64.36 | 51.64 | 58.6 | 53.19 | 80.58 |
| Baseline | 43.58 | 36.92 | 37.38 | 63.65 | 62.0 | 54.99 | 56.93 | 80.83 |
| Turn Aware Baseline | 43.08 | 38.5 | 38.53 | 66.96 | 64.62 | 57.31 | **58.99** | 81.58 |
| Turn Aware Speaker Graph | 44.53 | 39.11 | **40.06** | 66.32 | 63.72 | 57.36 | 58.81 | 80.86 |

Table 3: F1 score of the speaker-enriched model compared to prior work. The input to both the last two rows is augmented with turn tokens. ♣ are the results from rerunning the systems on our data and ♢ are our reimplementation.

Since some of these works use older, non-contextualized word embeddings such as GloVe (Pennington et al., 2014), we swap them with RoBERTa (Liu et al., 2019) to be comparable with our work and to make sure the gains in our approach aren't due to the use of a better language model. For a fair comparison across models, we also chunk the dialogues into same size. All results are an average of 5 random seeds. Details on changes made to any baselines to make them compatible with our data can be found in Appendix B.

**SwDA**  Adding graph-learned speaker representation does not help and brings the performance down slightly in terms of both F1 and accuracy. The speaker graph model is 0.18 F1 worse than the best-performing turn-aware baseline. SwDA is a two-party dataset and we postulate that adding speaker information in the form of a token to the input is better able to make use of the speaker turns. This may be because of inducing turn awareness at an earlier stage in the utterance before passing it to an RNN to capture sequential context.

On the other hand, modeling speakers using a graph has the downside of losing this sequential information (Ishiwatari et al., 2020). This could deteriorate the performance in dyadic dialogues where a speaker switch indicating token is enough to instill interlocutor information.

As mentioned in the earlier section, we mainly report and study the macro F1 scores to evaluate the models. We also present the accuracy in Table 3 for the sake of comparison with prior work.

**MRDA**  The turn-aware graph model gives the best performance on MRDA. When we compare the speaker graph model with the baseline where the input to both also contains the turn tokens, we see an improvement of 1.53 F1 score. For the rest of the discussion, we focus on MRDA, since we are interested in a multiparty dialogue setting.

### 6.2 Classwise Results

In this section, we analyze how the addition of speaker representations affects individual DA types. The detailed results can be found in Table 8. Performance on floor mechanisms improves with the addition of speaker representations. Floor mechanism tags all share a very similar vocabulary (Dhillon et al., 2004). In *hold*, a speaker is passed the floor while in *floor grabber*, the speaker tries to gain the floor. In order to disambiguate between these tags, it is vital to have information about who is speaking.

Without looking at prior turns, tags like *affirmative answer* and *negative response* can easily be confused with *statements* (Dhillon et al., 2004). These types of responses are hard to catch because disambiguating them requires analyzing the utterance in light of the dialogue context. The turn-aware speaker graph model gets a boost of 2.4 F1 points on *affirmative answer* and 5.4 F1 points on *negative*

| spk→utt | utt→utt | utt→spk | spk↔spk | E | F1 |
|:---:|:---:|:---:|:---:|:---:|:---:|
| ✓ | | | | $\|\mathcal{D}\|$ | 39.65 |
| ✓ | | | ✓ | $\|\mathcal{D}\| + \|\mathcal{P}\|$ | 39.57 |
| ✓ | | ✓ | | $2\|\mathcal{D}\|$ | 40.06 |
| ✓ | ✓ | | | $\|\mathcal{D}\|(1 + 2w)$ | 39.72 |
| ✓ | ✓ | ✓ | | $2\|\mathcal{D}\|(1 + w)$ | 39.89 |
| ✓ | ✓ | ✓ | ✓ | $2\|\mathcal{D}\|(1 + w) + 2\|\mathcal{P}\|$ | 39.43 |

Table 4: The effect of introducing different edge connections on the macro F1 score. E is the approximate number of total edges in the model. The results are sorted by increasing density of the graph in ascending order.

*answer*. This highlights the ability of the graph model to capture the situated speaker behavior.

**Performance on minority classes**    35 of the 50 classes in MRDA comprise less than 1% each of the data. Due to this class imbalance, it is difficult to build a system that performs well on these classes. The speaker graph model is able to make improvements on many infrequent classes. Some examples include *mimic, about-task, self-correct misspeaking, follow me, downplayer*.

The graph model can capture the subtle nuances of speaker behavior. Tags such as *mimic, downplayer, joke* see an improvement of 6.6, 3.8, 2.2 F1 score respectively. *mimic* utterances are those where the speaker repeats another speaker. They often serve as a form of acknowledgment from listeners (Dhillon et al., 2004). This shows certain intentions cannot be recovered from their semantic content alone.

### 6.3    Ablation Study on the Graph

In this section we study how introducing different edges affects the model's ability to capture speakers. First, we reiterate the edges used in our experiments along with introducing two new edge types.

- $(v_i^u, v_{S(u_i)}^s)$: A directed edge between an utterance and its speaker.
- $(v_i^u, v_j^u)$: A directed edge from one utterance to another.
- $(v_i^s, v_j^s)$: An undirected edge between two speakers.

To also capture context at the graph level, we connect utterances with each other. Having an edge from every utterance to every other is not feasible due to GPU memory limitations. Following Ghosal et al. (2019), every utterance node is connected with its immediate past and future utterances falling within a window $w$. That is, $v_i^u$ has edges of type $(v_i^u, v_j^u)$ to nodes $[v_{i-w}^u, .., v_{i-1}^u, v_i^u, v_{i+1}^u, v_{i+w}^u]$. In all our experiments, $w$ is set to 10.

It is clear from Table 4 that the most important edge type is the one connecting every speaker with its own utterances. Capturing context at the graph level through the $utt \rightarrow utt$ edge is not needed to model speakers well and hurts the performance slightly. One possible explanation for this is that the utterance nodes are already initialized with contextualized utterance vectors. Furthermore, connecting speakers with each other ($spk \leftrightarrow spk$) hurts the performance in all cases. This shows that giving the speaker nodes direct access to only their own utterances is important and adding more edges can potentially introduce noise. The simplest graph with just a single type of edge ($v_i^u, v_{S(u_i)}^s$) denoted by $spk \rightarrow utt$ results in an F1 score that is still 1 point better than the turn-aware baseline.

The best-performing model also has edges between the utterances and their respective speakers, i.e., the $(v_i^u, v_{S(u_i)}^s)$ edges becomes undirected. Any $(v_i^u, v_j^u)$ edges introduced to this setting hurts the performance. Capturing context at the graph level becomes computationally expensive and often impossible as the dialogue size increases. One common workaround is to connect utterances falling within a window size of each other. Even in such a setting, the memory requirements incurred are often large and inhibit the value of $w$ to be set too big. We show in our work that this is not only redundant but also hurts the performance. Instead, a simple objective of learning speaker representations directly is not only compute efficient but performs better as well. The best performing model has $2|\mathcal{D}|$ edges (row three of Table 4), whereas any graph with $utt \rightarrow utt$ edges introduces an additional $2|\mathcal{D}|w$ edges. While both settings have the number of edges as a linear function of the dialogue length, in practice the former can be more effective in low-resource settings such as under GPU memory and speed constraints.

| Chunk Size | Max$|\mathcal{P}|$ | Avg$|\mathcal{P}|$ | F1 |
|---|---|---|---|
| Full | 9 | 6.13 | – |
| 128 | 8 | 5 | 40.06 |
| 96 | 8 | 4.8 | 39.96 |
| 64 | 8 | 4.5 | 39.82 |
| 32 | 8 | 4 | 39.63 |
| 16 | 7 | 3.3 | 39.62 |

Table 5: Segmenting MRDA dialogues into smaller chunks of various sizes results in fewer active speakers in that chunk. Here $|\mathcal{P}|$ is the number of participants in the dialogue.

## 6.4 Effect of Chunk Size

Modeling long conversations in one go is not feasible due to computational resource limitations. Therefore, we segment the dialogues into smaller, more manageable chunks. The maximum and average number of speakers corresponding to each chunk size along with the macro F1 scores of the graph model are shown in Table 5. It is possible that this segmentation can strip away useful information in cases where an utterance at a later stage of the dialogue relies on an earlier utterance to resolve context issues.

Our experiments show there is not much difference in performance by choosing a smaller chunk size. A smaller chunk size of 16 still gives 1 macro F1 score improvement over the turn-aware baseline that had access to a larger context of 128. Furthermore, we observed performance gains even when only 3 speakers on average are involved in a multiparty setting, as opposed to the purely dyadic case of SwDA where no gains were observed. This highlights the usefulness of speaker modeling in multi-participant dialogues. The results on the validation set are presented in Table 7.

## 6.5 Utterance Graph Nodes

In this section we present the results of the model after using the utterance nodes from the graph instead of the speaker nodes to augment the utterance representations from Equation 1. The $h_i^{\text{speaker-enriched}}$ becomes:

$$h_i^{\text{speaker-enriched}} = [h_i; \widetilde{v}_i^u] \qquad (7)$$

Similarly, we also include the results of concatenating both speaker and utterance nodes. Table 6 shows that although all three systems perform better than the baseline, the model with speaker nodes

| Graph Node | Macro F1 |
|---|---|
| Speaker | 40.06 |
| Utterance | 39.46 |
| Both speaker and utterance | 39.47 |

Table 6: The results on MRDA by concatenating graph-learned speaker, utterance, and both speaker and utterance nodes with the utterance representations from the sequential encoder.

concatenated performs the best. Our assumption is that since the model contains two GNN layers, the utterance nodes are also able to implicitly learn more context through the edge that connects them with their speaker. Context in this case is the utterances by the same speaker because utterances from different speakers do not interact at the graph level.

## 7 Conclusion

We propose a graph-based approach to learn speaker-informed utterance representations for DA classification. We show that directly learning speaker representations with a simple graph is both effective and efficient. Instilling speaker information this way helps disambiguate DA labels in the case of multiparty dialogues. The learned speaker representations can be used on top of any utterance encoding scheme to include speaker information. Future work can also look into incorporating audio features to encode a speaker since prosody plays an important part in signaling intention.

## Acknowledgements

We would like to thank the anonymous reviewers for their valuable feedback and input. Research was sponsored by the Army Research Office and was accomplished under Grant Number W911NF-20-1-0002. The views and conclusions contained in this document are those of the authors and should not be interpreted as representing the official policies, either expressed or implied, of the Army Research Office or the U.S. Government. The U.S. Government is authorized to reproduce and distribute reprints for Government purposes notwithstanding any copyright notation herein.

## Limitations

Training a graph neural network with longer dialogues and dense edges can be infeasible because of GPU memory restrictions. For example, the choice

of type of edge connections to have along with other hyperparameters such as the window size can be restrictive because of compute resource limitations.

## Ethics Statement

One potential use of dialogue act recognition is in the domain of conversational agents and chatbots. AI assistants can be used to carry out commands for users. If the underlying DA recognition systems used by such assistants are brittle, they can negatively impact the end users. Since the performance of DA systems on finer-grained labels is not robust, caution should be used while integrating them with real-world applications.

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

## A  Experimental Setup

A learning rate (LR) scheduler reduces the LR by 0.1 after 4 epochs of no improvement on the validation set. We train the models, including all the baselines, for a maximum of 100 epochs with early stopping after no improvement on the macro F1 on the validation set for 10 epochs.

For training the baseline encoder, we keep most of the hyperparameters the same as He et al. (2021). The GRU hidden size used for the encoder is 200, since this is the largest encoder model we could fit on the GPU due to memory constraints when using the output from the encoder as input to the graph model.

The window hyperparameter $w$ was chosen from a search space of $\{0, 5, 10\}$. There are two layers of RGAT with attention computed "across-relation" using "additive-self-attention". The number of attention heads for both layers is set to two.

The graph model on top of the encoder has around 154M parameters. All the experiments on MRDA are done on a single NVIDIA A100 GPU, and the model converges in about half an hour. The SwDA experiments are all done on a single NVIDIA RTX A6000, with a single run taking approximately an hour.

## B  Baseline Models

All the systems included (except DialogXL) use RoBERTa-base as the pre-trained language model. For all the graph models, the hyperparameter $w$ is set to 10–the same as in our graph model.

- We make two changes to SUNET (Song et al., 2023) in the results we report after replicating their system. First, They use the utterance representations of all the utterances of a speaker from the training set to initialize that speaker node. We randomly initialize speakers. The utterance nodes are initilized using trained encoder from § 5.3. This was because we couldn't fit the model on GPU by directly using RoBERTa representations since those have a large dimension compared to the encoder (768 vs 400).

| Chunk Size | F1 |
|:----------:|:----:|
| Full | – |
| 128 | 38.58 |
| 96 | 38.63 |
| 64 | 38.62 |
| 32 | 38.59 |
| 16 | 38.29 |

Table 7: Effect of chunk size on the validation set of MRDA for the speaker graph model.

- For Turn Modeling (He et al., 2021) system, we do not include *Topic* embeddings for the SwDA dataset.

## C   Results on Validation set

We present the results on the validation set here. All the results are an average of 5 runs. Even though the model with chunk size 96 performs slightly better on the validations set, We picked the chunk size to be 128. This makes for a fair comparison with the baseline that uses 128-sized chunks.

## D   Classwise Results

The test set does not have any sample of the *welcome* class.

| % Distribution | Label | Description | Turn-aware Baseline | Baseline | Turn-aware Speaker Graph |
|---|---|---|---|---|---|
| 34.73 | s | Statement | 73.2 | 76.8 | 76.2 |
| 1.29 | qw | Wh-Question | 81.2 | 81.2 | 80.6 |
| 0.24 | qh | Rhetorical Question | 9.6 | 9.0 | 14.6 |
| 0.35 | qrr | Or Clause After Y/N Question | 87.6 | 88.8 | 90.0 |
| 0.15 | qr | Or Question | 63.8 | 66.2 | 64.2 |
| 0.11 | qo | Open-ended Question | 18.0 | 13.0 | 16.20 |
| 1.97 | qy | Y/N Question | 67.8 | 68.8 | 68.2 |
| 0.79 | g | Tag Question | 72.0 | 72.8 | 72.2 |
| 5.47 | aa | Accept | 48.0 | 50.6 | 51.4 |
| 0.23 | aap | Partial Accept | 5.8 | 3.2 | 5.6 |
| 1.08 | na | Affirmative Answer | 14.4 | 15.2 | 17.6 |
| 0.87 | ar | Reject | 78.4 | 78.6 | 78.8 |
| 0.14 | arp | Partial Reject | 0.0 | 0.0 | 2.0 |
| 0.48 | nd | Dispreferred Answer | 16.0 | 15.8 | 16.4 |
| 0.34 | ng | Negative Answer | 2.4 | 2.2 | 7.8 |
| 14.02 | b | Backchannel | 68.2 | 78.6 | 79.0 |
| 6.74 | bk | Acknowledgement | 56.8 | 59.4 | 58.2 |
| 2.12 | ba | Assessment/Appreciation | 58.2 | 58.2 | 58.2 |
| 0.14 | bh | Rhetorical Question Backchannel | 52.0 | 57.2 | 65.2 |
| 0.24 | h | Hold | 36.4 | 50.8 | 53.6 |
| 5.77 | fh | Floor Holder | 80.0 | 81.8 | 82.0 |
| 1.3 | fg | Floor Grabber | 7.8 | 11.8 | 20.0 |
| 0.38 | cc | Commitment | 37.8 | 35.4 | 33.0 |
| 2.65 | cs | Suggestion | 46.8 | 46.0 | 45.8 |
| 0.7 | co | Command | 53.0 | 54.2 | 51.0 |
| 0.33 | am | Maybe | 44.4 | 46.4 | 45.6 |
| 0.82 | no | No Knowledge | 68.0 | 65.6 | 67.0 |
| 0.27 | m | Mimic | 2.4 | 2.4 | 9.0 |
| 0.2 | r | Repeat | 0.0 | 0.0 | 0.0 |
| 0.14 | bs | Summary | 0.0 | 0.0 | 0.0 |
| 0.59 | f | Follow Me | 39.8 | 41.0 | 47.0 |
| 0.22 | br | Repetition Request | 61.0 | 58.6 | 60.6 |
| 1.99 | bu | Understanding Check | 46.4 | 49.8 | 48.0 |
| 3.55 | df | Defending/Explanation | 48.6 | 49.4 | 46.4 |
| 3.06 | e | Elaboration | 31.2 | 36.6 | 36.4 |
| 0.78 | 2 | Collaborative Completion | 13.0 | 25.8 | 25.2 |
| 0.05 | bc | Correct Misspeaking | 0.0 | 0.0 | 0.0 |
| 0.14 | bsc | Self-Correct Misspeaking | 1.8 | 0.0 | 14.2 |
| 0.38 | bd | Downplayer | 4.4 | 6.6 | 10.4 |
| 0.01 | by | Sympathy | 0.0 | 0.0 | 0.0 |
| 0.24 | fa | Apology | 84.6 | 85.8 | 88.4 |
| 38.0 | ft | Thanks | 86.8 | 84.8 | 86.4 |
| 0.01 | % | Welcome | - | - | - |
| 0.28 | fe | Exclamation | 81.0 | 80.4 | 80.6 |
| 0.4 | t | About-Task | 9.6 | 6.4 | 12.2 |
| 0.29 | tc | Topic Change | 5.2 | 5.0 | 7.2 |
| 0.44 | j | Joke | 4.8 | 0.6 | 7.0 |
| 0.21 | t1 | Self Talk | 9.0 | 8.2 | 5.4 |
| 0.17 | t3 | Third Party Talk | 0.0 | 0.0 | 0.0 |
| 3.03 | % | Disruption | 48.4 | 57.0 | 59.4 |

Table 8: Per class macro F1 scores on the test set for all models on MRDA. Distribution is the percentage of labels of a class in the whole data. Scores are the average of 5 random runs.