# OpenReview forum: "Who is Speaking? Speaker-Aware Multiparty Dialogue Act Classification"
_EMNLP/2023/Conference — EMNLP 2023 Findings_

### Official Review · Reviewer_tXBt · 2023-08-04

**Soundness:** 3

**Excitement:**

3: Ambivalent: It has merits (e.g., it reports state-of-the-art results, the idea is nice), but there are key weaknesses (e.g., it describes incremental work), and it can significantly benefit from another round of revision. However, I won't object to accepting it if my co-reviewers champion it.

**Paper Topic And Main Contributions:**

The paper proposed a GNN-based approach to dialog act recognition.

The new model mainly consists of three components
- A speaker turn token is prepended to each utterance following past works.
- An utterance encoder that uses RoBERTa outputs as utterance embeddings.
- A graph neural network that maintains speaker nodes and utterance nodes. utterance-utterance and speaker-utterance edges are constructed to propagate information.
And the updated GNN node embeddings are used for the final classification.

Experimental evaluation on MRDA and SwDA shows that the speaker turn token contributes to the most improvement. The GNN component is complex but has very limited effect. In some cases adding the GNN even degrades the performance.

**Reasons To Accept:**

RA 1: It proposed to use GNN to encode speaker information.

**Reasons To Reject:**

RR 1: Limited effect of the proposal.

RR 2: The minority class analysis in Section 6.2 does not consider sample size. For example, MRDA data has 15K test utterances, *mimic* and *repeat* consisting of 15Kx0.27%=41 and 15Kx0.2%=30 samples, respectively. The improved 4.8 and 8.6 F1 scores will **roughly** correspond to 41*4.8%=2 and 30*8.6%=2.6 data samples. I suggest the authors provide significance test results to support their conclusions.

**Reproducibility:**

2: Would be hard pressed to reproduce the results. The contribution depends on data that are simply not available outside the author's institution or consortium; not enough details are provided.

**Reviewer Confidence:**

3: Pretty sure, but there's a chance I missed something. Although I have a good feel for this area in general, I did not carefully check the paper's details, e.g., the math, experimental design, or novelty.

**Typos Grammar Style And Presentation Improvements:**

Important results in the main body should use support from the main body, but at L466 the authors refer to results in the Appendix.

---

> ### Author Rebuttal · Authors · 2023-08-29
>
> Thank you for the review and your suggestions.
>
> > RR 1: Limited effect of the proposal.
>
> We have worked under the fine-grained setting of DA labels as opposed to only the top level that prior work has focused on. Fine-grained DA labels are important for many downstream applications but they are also more difficult to gain performance boosts on because of the extremely skewed data distribution and the inherent difficulty of the task. Therefore, gaining improvements on this task is not easy.
>
> ---
>
> > RR 2.
>
> We conducted Student’s t-test and our model’s performance is statistically significant with p<0.0001.

---

### Official Review · Reviewer_pYXY · 2023-08-04

**Soundness:** 2

**Excitement:**

2: Mediocre: This paper makes marginal contributions (vs non-contemporaneous work), so I would rather not see it in the conference.

**Paper Topic And Main Contributions:**

The paper aims to use explicit speaker representations to improve the performance of dialogue act classification. They propose a GNN based network to capture speaker specific nuances and Transformer based model to encode utterances.

**Reasons To Accept:**

- Relevant approach to DA classification.
- Relevant use of graphs to monitor speakers in dialogue.
- Modular approach provided which can be used with other utterance encoders as well.

**Reasons To Reject:**

- Very loose papers with many unnecessary details.
- Idea and approach not novel.
- Insufficient comparative systems used.


**Reproducibility:**

3: Could reproduce the results with some difficulty. The settings of parameters are underspecified or subjectively determined; the training/evaluation data are not widely available.

**Reviewer Confidence:**

4: Quite sure. I tried to check the important points carefully. It's unlikely, though conceivable, that I missed something that should affect my ratings.

---

> ### Author Rebuttal · Authors · 2023-08-29
>
> Thank you for your review.
>
> > Very loose papers with many unnecessary details.
>
> We believe that it is in the interest of reproducibility to err on the side of providing too much detail, rather than too little. However, we are open to any constructive feedback you may have on which details are not necessary.
>
> ---
>
> > Idea and approach not novel.
>
> We argue in our paper that explicitly modeling speakers for DA classification is important. While prior work for DA has, to a limited extent, tried to model speakers, the approaches are limited to two-party conversations and also don’t scale well to unseen speakers at test time. A thorough discussion can also be found in the paper L052 - L069 and L102-L131. The approach of using graph neural networks to encode a dialogue and inject speakers has been presented before, building on that work we create a simple GNN with utterance and speaker nodes. We explicitly optimize for learning good speaker representations in contrast to using GNNs to encode more utterances’ context, as is done in most prior work (L161-L169). For a more detailed discussion please also refer to the section “Graph-based methods to encode speakers in dialogues”.
>
> ---
>
> > Insufficient comparative systems used
>
> We have run another system from the field of emotion recognition that incorporates speaker information. The results on MRDA are shown below. Results are average on 3 random runs after an extensive hyperparameter search.
>
> | Model      | Macro F1 |
> | :---        |    :----:   |
> | Our baseline        |    38.69  |
> | Our approach        |    39.89  |
> | DAG-ERC [1]   |  25.15        |
>
> [1] build a directed acyclic graph (DAG) with utterances as the only nodes. They incorporate the speaker information by distinguishing the edge types one utterance can have with another: one for when the utterances are by the same speaker and another edge type if they are spoken by different speakers. There is no explicit learning of speaker representations. They only use two different edge types to indicate if two connected utterances share the same speaker or not. This technique does not work well for multiparty dialogues as it essentially converts it to a two-party conversation.
>
> ---
>
> References
>
> [1] Shen, W., Wu, S., Yang, Y., & Quan, X. (2021). Directed acyclic graph network for conversational emotion recognition. arXiv preprint arXiv:2105.12907.

---

### Official Review · Reviewer_JZjf · 2023-08-09

**Soundness:** 3

**Excitement:**

4: Strong: This paper deepens the understanding of some phenomenon or lowers the barriers to an existing research direction.

**Paper Topic And Main Contributions:**

This paper presents a new approach to dialogue act (DA) classification using a graph neural network (GNN) and turn-indicating tokens. They demonstrate on the fine-grained DA prediction task from the MRDA dataset that speaker representations learned from the GNN outperform other baselines, specifically in multi-party dialogues. The paper itself is clear and well-written, with a thorough literature review that makes it easy for the reader to see how their work differs from prior work. The explanation in the analysis section for why the graph helps with multi-party dialogues seems reasonable. The GNN approach for DA classification is interesting and novel.



**Questions For The Authors:**

Question A: Is the improvement in F1 on MRDA significant?

**Reasons To Accept:**

The paper itself is clear and well-written, with a thorough literature review that makes it easy for the reader to see how their work differs from prior work. The explanation in the analysis section for why the graph helps with multi-party dialogues seems reasonable. The GNN approach for DA classification is interesting and novel.



**Reasons To Reject:**

It does not appear that the code has been made open-source. The results look good, although they are not completely convincing since most of the similar approaches mentioned in the related work (Chi, Colombo, Ghosal, Zhang, or Sun) are not included, it is unclear whether the improvement in F1 score on MRDA is significant since the increase is small, and the graph does not help for SwDA.

**Reproducibility:**

4: Could mostly reproduce the results, but there may be some variation because of sample variance or minor variations in their interpretation of the protocol or method.

**Reviewer Confidence:**

3: Pretty sure, but there's a chance I missed something. Although I have a good feel for this area in general, I did not carefully check the paper's details, e.g., the math, experimental design, or novelty.

**Typos Grammar Style And Presentation Improvements:**

•	It’s confusing why the graph structure subsection in 4.3 is included when some of those relations do not match the edges in Figure 1. It seems like the graph construction subsection is clearer and sufficient. OR explicitly state which are not in the figure and why.
•	Explain a bit about what the RGAT function actually does.
•	Table 2 appears much later than when it’s first referenced in the text.
•	When Table 3 is referenced, accuracy is mentioned, so you should reference Table 4 too.
•	In footnote 6, you mention that the hidden size is 200. Why not use 384?
•	“into same size” -> “into the same size”
•	“are average” -> “are the average”
•	In the column headings in Table 5, what is the italicized P?

---

> ### Author Rebuttal · Authors · 2023-08-29
>
> Thank you for your review and suggestions.
>
> > It does not appear that the code has been made open-source.
>
> We will open-source our code with the camera-ready version.
>
> ---
>
> > similar approaches mentioned in the related work (Chi, Colombo, Ghosal, Zhang, or Sun) are not included
>
>  All these papers do not have their code publicly available (except for Ghosal, those results are included in the paper), which makes comparing with them difficult. Here we present an additional relevant system from emotion recognition on MRDA dataset. Results are average on 3 random runs after an extensive hyperparameter search.
>
> | Model      | Macro F1 |
> | :---        |    :----:   |
> | Our baseline        |    38.69  |
> | Our approach        |    39.89  |
> | DAG-ERC [1]   |  25.15        |
>
> [1] build a directed acyclic graph (DAG) with utterances as the only nodes. They incorporate the speaker information by distinguishing the edge types one utterance can have with another: one for when the utterances are by the same speaker and another edge type if they are spoken by different speakers. There is no explicit learning of speaker representations. They only use two different edge types to indicate if two connected utterances share the same speaker or not. This technique does not work well for multiparty dialogues as it essentially converts it to a two-party conversation.
>
>
> ---
>
> > Question A: Is the improvement in F1 on MRDA significant?
>
> Yes. We conducted Student’s t-test and our model’s performance is statistically significant with p<0.0001.
>
> ---
>
> > In footnote 6, you mention that the hidden size is 200. Why not use 384?
>
> The output from the utterance encoder is fed to the graph neural network. Setting the hidden size to 384 resulted in utterance representations of size 768 since these utterance representations were the input to the graph, we ran into a cuda out of memory error. So we had to restrict the hidden size to 200 to fit the graph on gpu.
>
> ---
>
> References
>
> [1] Shen, W., Wu, S., Yang, Y., & Quan, X. (2021). Directed acyclic graph network for conversational emotion recognition. arXiv preprint arXiv:2105.12907.

---

### Official Review · Reviewer_U5G3 · 2023-08-09

**Soundness:** 2

**Excitement:**

2: Mediocre: This paper makes marginal contributions (vs non-contemporaneous work), so I would rather not see it in the conference.

**Missing References:**

[1] Song, Rui, et al. "SUNET: Speaker-utterance interaction Graph Neural Network for Emotion Recognition in Conversations." Engineering Applications of Artificial Intelligence 123 (2023): 106315.
[2] Shen, Weizhou, et al. "Directed Acyclic Graph Network for Conversational Emotion Recognition." Proceedings of the 59th Annual Meeting of the Association for Computational Linguistics and the 11th International Joint Conference on Natural Language Processing (Volume 1: Long Papers). 2021.
[3] Lee, Bongseok, and Yong Suk Choi. "Graph Based Network with Contextualized Representations of Turns in Dialogue." 2021 Conference on Empirical Methods in Natural Language Processing (EMNLP 2021). ASSOC COMPUTATIONAL LINGUISTICS-ACL, 2021.
[4] Liang, Yunlong, et al. "Infusing multi-source knowledge with heterogeneous graph neural network for emotional conversation generation." Proceedings of the AAAI Conference on Artificial Intelligence. Vol. 35. No. 15. 2021.
[5] Ishiwatari, Taichi, et al. "Relation-aware graph attention networks with relational position encodings for emotion recognition in conversations." Proceedings of the 2020 Conference on Empirical Methods in Natural Language Processing (EMNLP). 2020.

**Paper Topic And Main Contributions:**

In this paper, the authors consider introducing speakers' features in the process of Dialogue acts (DAs) recognition and modeling multi-round conversations involving multiple speakers by constructing heterogeneous graphs and learning conversations containing speakers' features using RGAT to facilitate classification. The authors emphasize that there has been little previous work on incorporating speaker awareness into DA classification.

**Questions For The Authors:**

No.

**Reasons To Accept:**

1. Very well written and logical.
2. Detailed parameter exploration and rich experimental details.
3. Sufficient experimental datasets.

**Reasons To Reject:**

1. Insufficient innovation. Although the application scenario of this paper is DA classification, the authors still need to explore and compare the related studies on Emotion Recognition in Conversations (ERC) because the two fields are too similar. As a point of comparison, the authors' main technical details, both the construction of heterogeneous graphs and the introduction of speakers' information into the dialog, have been heavily discussed in [1, 2, 3, 4]. In addition, the use of RGAT is also not new [5]. Therefore, the authors cannot substantiate the innovativeness of the proposed method without discussing the above work.
2. Too weak baselines. As articulated in the previous point, the authors need to make a more comprehensive experimental comparison of similar methods, but the manuscript only shows DialogueGCN, DialogueRNN and appraoch in [6]. I don't doubt for a second that recent work in the field can achieve results similar to or better than the authors' proposed method.

[1] Song, Rui, et al. "SUNET: Speaker-utterance interaction Graph Neural Network for Emotion Recognition in Conversations." Engineering Applications of Artificial Intelligence 123 (2023): 106315.
[2] Shen, Weizhou, et al. "Directed Acyclic Graph Network for Conversational Emotion Recognition." Proceedings of the 59th Annual Meeting of the Association for Computational Linguistics and the 11th International Joint Conference on Natural Language Processing (Volume 1: Long Papers). 2021.
[3] Lee, Bongseok, and Yong Suk Choi. "Graph Based Network with Contextualized Representations of Turns in Dialogue." 2021 Conference on Empirical Methods in Natural Language Processing (EMNLP 2021). ASSOC COMPUTATIONAL LINGUISTICS-ACL, 2021.
[4] Liang, Yunlong, et al. "Infusing multi-source knowledge with heterogeneous graph neural network for emotional conversation generation." Proceedings of the AAAI Conference on Artificial Intelligence. Vol. 35. No. 15. 2021.
[5] Ishiwatari, Taichi, et al. "Relation-aware graph attention networks with relational position encodings for emotion recognition in conversations." Proceedings of the 2020 Conference on Empirical Methods in Natural Language Processing (EMNLP). 2020.
[6] He, Zihao, et al. "Speaker Turn Modeling for Dialogue Act Classification." Findings of the Association for Computational Linguistics: EMNLP 2021. 2021.

**Reproducibility:**

4: Could mostly reproduce the results, but there may be some variation because of sample variance or minor variations in their interpretation of the protocol or method.

**Reviewer Confidence:**

4: Quite sure. I tried to check the important points carefully. It's unlikely, though conceivable, that I missed something that should affect my ratings.

---

> ### Author Rebuttal · Authors · 2023-08-29
>
> Thank you for your review and for bringing the related work to our attention.
>
> >“Insufficient innovation”
>
> We briefly summarize and compare the approaches in [1-5] to our work.
> [1] This is a very recent work (published < 3 months ago) that also incorporates speaker information through GNNs. They update the speaker representation through a different update function than ours and they also include regularization loss to discourage similarity between different speaker representations. They construct the graph using three types of edges. One that connects every utterance to its speaker, one that connects an utterance with the utterances by the same speaker, and the last edge connects utterances with utterances of other speakers. We build the graph a little differently-- utterances are connected with a single type of edge.
> [2] build a directed acyclic graph (DAG) where the nodes are the utterances. They argue that a DAG is more suitable for dialogue modeling because information cannot flow backward from future utterances to the past. They incorporate the speaker information by distinguishing the edge types one utterance can have with another: one for when the utterances are by the same speaker and another edge type if they are spoken by different speakers. There are multiple differences between this work and ours. The main difference is the way they construct the graph---there is no explicit learning of speaker representations. They only use two different edge types to indicate if two connected utterances share the same speaker or not. In some ways, this is akin to converting a multiparty dialogue to a two-party because the utterances only have binary information of speakers now. This is not an appropriate way to handle multiparty dialogues (a conclusion also mentioned in [2]).
> [3] Use BERT (or its variants) to encode utterances, followed by a turn attention module and an RNN to inject context and sequential information respectively. Then a dialogue graph is constructed. They treat ERC as a link prediction task between the speaker and the utterance node. Speaker information is encoded in two ways. (1) A speaker token id is summed with each token’s representation from a BERT/BERT-like model. This requires learning a fixed number of speaker embeddings. (2) In the dialogue graph, each utterance of the same speaker is connected. Again, doing so does not explicitly learn speaker representations for each dialogue.
> [4] Present a way to incorporate emotion and speaker traits for emotional conversation generation. They construct a graph with utterance, speaker, emotion, and facial expression nodes. The speaker nodes are learned through a matrix of fixed size (the number of unique speakers in data). They experimented with a dataset derived from a TV show with a fixed number of dominant speakers. On the contrary, we target the setting of natural meetings where we assume that at test time we could be given new speakers not seen during training.
> [5] Incorporate sequential information in relational graph attention network (RGAT) [7] through positional embeddings. The nodes in their graph are only utterances and they model speaker dependency by four different types of edges between utterances: whether utterances are from the same speaker or not combined with directionality of past and future. This technique suffers the same pitfalls of not scaling to multiparty conversations. In contrast, we add speaker nodes to the graph.
>
> ---
>
> >Too weak baselines.
>
> Below we show results after rerunning the system from [2]. The results are on MRDA dataset which is the primary dataset we experiment with. We conducted an extensive hyperparameters search and report the final results from the epoch with best validation F1. We used learning rate from [ 5e-2, 1e-3, 2e-3, 3e-3, 4e-3,  5e-3, 9e-4, 5e-4, 1e-4, 5e-5 ], batch size from [64, 128, 256] and trained for a maximum of 300 epochs.
>
>
> | Model      | Macro F1 |
> | :---        |    :----:   |
> | Our baseline        |    38.69  |
> | Our approach        |    39.89  |
> | DialogueGCN [8]      | 34.67       |
> | DAG-ERC [2]   |  25.15        |
>
>
> Multiparty DA classification under the fine-grained setting is a challenging task. DAG-ERC performs poorly on MRDA. Our assumption is that DAG-ERC is more suitable for two-party dialogues. Other systems we have compared with in the paper from ERC [8] also perform poorly than our baseline.
>
>
>
> ---
>
> References:
>
> [7] Busbridge, D., Sherburn, D., Cavallo, P., & Hammerla, N. Y. (2019). Relational graph attention networks. arXiv preprint arXiv:1904.05811.
> [8] Ghosal, D., Majumder, N., Poria, S., Chhaya, N., & Gelbukh, A. (2019). Dialoguegcn: A graph convolutional neural network for emotion recognition in conversation. arXiv preprint arXiv:1908.11540.

---

### Meta-Review · Area_Chair_WjPw · 2023-09-17

**Recommendation:** 2

**Metareview:**

This paper studies the role of speaker information in the task of dialog act classification. It proposes to use speaker-information using graph neural networks to improve the utterance representations. Experiments on both dyadic and multi-participant conversations demonstrate the utility of the proposed model.

Overall, the paper, while well-written, suffers some concerns with respect to novelty. In particular, discussions amongst reviewers and authors revealed the existence of many related approaches that share the same motivation – using/encoding speaker information using some form of graph network. Given minor differences, strong justification for novelty was missing.

Another valid suggestion revolves around space optimization where space could be used to perform a better comparison to graph-based baselines and also non-GAT-based speaker-encoding baselines.

---

### Decision · Program_Chairs · 2023-10-07

**Decision:**

Accept-Findings

**Comment:**

This paper studies the role of speaker information in the task of dialog act classification. It proposes to use speaker-information using graph neural networks to improve the utterance representations. Experiments on both dyadic and multi-participant conversations demonstrate the utility of the proposed model.

Overall, the paper, while well-written, suffers some concerns with respect to novelty. In particular, discussions amongst reviewers and authors revealed the existence of many related approaches that share the same motivation – using/encoding speaker information using some form of graph network. Given minor differences, strong justification for novelty was missing.

Another valid suggestion revolves around space optimization where space could be used to perform a better comparison to graph-based baselines and also non-GAT-based speaker-encoding baselines.